# Clinical Outcomes and Prognostic Factors for Extramammary Paget’s Disease Treated with Radiation Therapy: A Multi-Institutional Observational Study

**DOI:** 10.3390/cancers17091507

**Published:** 2025-04-29

**Authors:** Masanari Niwa, Natsuo Tomita, Hiromichi Ishiyama, Hijiri Kaneko, Yukihiko Oshima, Hirota Takano, Masayuki Matsuo, Mayu Kuno, Akifumi Miyakawa, Shinya Otsuka, Taiki Takaoka, Dai Okazaki, Akira Torii, Nozomi Kita, Seiya Takano, Motoki Nakamura, Hiroshi Kato, Akimichi Morita, Akio Hiwatashi

**Affiliations:** 1Department of Radiology, Graduate School of Medical Sciences, Nagoya City University, 1 Kawasumi, Mizuho-cho, Mizuho-ku, Nagoya 467-8601, Japan; ntomita@med.nagoya-cu.ac.jp (N.T.); ahiwata@med.nagoya-cu.ac.jp (A.H.); 2Department of Radiation Oncology, Kitasato University Hospital, 1-15-1 Kitasato, Minami-ku, Sagamihara 252-0375, Japan; hishiyam@kitasato-u.ac.jp (H.I.);; 3Department of Radiology, Aichi Medical University Hospital, 1-1 Yazako-Karimata, Nagakute 480-1195, Japan; ooshima.yukihiko.884@mail.aichi-med-u.ac.jp; 4Department of Radiation Oncology, Gifu University Hospital, 1-1 Yanagido, Gifu 500-1194, Japanmatsuo.masayuki.e0@f.gifu-u.ac.jp (M.M.); 5Department of Radiation Oncology, Ichinomiya Municipal Hospital, 2-2-22, Bunkyo, Ichinomiya 491-8558, Japan; 6Department of Radiation Oncology, National Hospital Organization Nagoya Medical Center, 4-1-1, Sannomaru, Nakaku, Nagoya 460-0001, Japan; 7Department of Radiology, Okazaki City Hospital, 3-1, Kouryuuji-cho Gosyoai, Okazaki 444-8553, Japan; 8Department of Geriatric and Environmental Dermatology, Nagoya City University Graduate School of Medical Sciences, 1 Kawasumi, Mizuho-cho, Mizuho-ku, Nagoya 467-8601, Japanamorita@med.nagoya-cu.ac.jp (A.M.)

**Keywords:** radiation therapy, extramammary Paget’s disease, treatment outcome

## Abstract

Extramammary Paget’s disease (EMPD) is a rare skin cancer that usually affects older adults and appears in the genital and perianal regions. This study analyzed the outcomes of radiation therapy in 81 patients with EMPD. The three-year local control, progression-free survival, and overall survival rates were 75%, 52%, and 80%, respectively. The results showed that lymph node metastasis, not having surgery, and using local radiation only without treating nearby lymph nodes were linked to worse outcomes. Side effects were generally mild, with only one patient experiencing moderate skin infection and another having severe lymphedema, while no very severe side effects occurred. These findings suggest that surgery and treating nearby lymph nodes in addition to radiation may improve outcomes for EMPD patients. Further research is needed to determine the best treatment approach, especially for older patients with poor prognostic factors.

## 1. Introduction

Extramammary Paget’s disease (EMPD) is a rare cutaneous adenocarcinoma that primarily targets the genital and perianal regions, along with other areas rich in apocrine glands [1,2]. While Paget’s disease is almost exclusively diagnosed in women, with male cases accounting for less than 1% and occasionally linked to prostate cancer, invasive EMPD affects both sexes, but is more common in women [3]. Clinically, EMPD lesions frequently present with invasive redness accompanied by crusts and scaling, which sometimes resemble other skin conditions, such as eczema [4,5]. The majority of EMPD cases are typically identified as carcinoma in situ, and disease progression is generally slow. However, when Paget cells deeply infiltrate the dermis, regional lymph node (LN) metastases and distant metastases often develop and the prognosis of these patients is poor [6,7].

Although surgical excision remains the main curative approach for EMPD, its feasibility can be constrained in elderly patients, those with significant comorbidities, or individuals reluctant to undergo surgical interventions [7,8,9]. In such cases, radiation therapy (RT) is considered a viable alternative; however, clinical outcomes related to RT are underreported due to the limited number of patients receiving definitive RT [8,10]. Moreover, despite advancements in clinical outcomes for various cancers through the utilization of modern RT techniques, including intensity-modulated radiation therapy (IMRT) [11,12,13], there is a scarcity of studies documenting the application of these advanced techniques in EMPD management. Consequently, the current study aims to investigate the clinical outcomes of non-metastatic EMPD treated with RT, either as monotherapy or in conjunction with surgery and/or chemotherapy, while also identifying prognostic factors that could inform optimal treatment strategies.

## 2. Materials and Methods

### 2.1. Patient Selection and Pretreatment Evaluation

This retrospective, observational study was conducted with approval from the Institutional Review Board of our facility (approval number: 60-23-0073) and 10 other participating centers. The study followed the ethical guidelines set forth in the Declaration of Helsinki (1964) and its subsequent revisions. As this was a retrospective analysis, patient consent was acquired through an opt-out system available on the institutional website. We examined patients diagnosed with EMPD who received radiotherapy (RT), either as monotherapy or in combination with surgery and/or chemotherapy, between 2004 and 2022. Since this was a retrospective multi-institutional study, there were no strict criteria for selecting patients for RT. In general, RT was considered for patients with residual disease after surgery, those who were not surgical candidates due to advanced age or other factors. The inclusion criteria were as follows: (1) histopathologically confirmed EMPD; (2) the primary site of the lesion located outside of the breast; and (3) no evidence of distant metastasis. A total of 81 non-metastatic EMPD patients were enrolled in the study.

Clinical staging was determined based on physical examination findings and imaging studies, including computed tomography (CT) scans of the head, neck, chest, and upper abdomen. Additionally, magnetic resonance imaging (MRI) and 18F-fluorodeoxyglucose positron emission tomography (PET)/CT were performed on 6 (7%) and 18 (22%) patients, respectively. Tumor thickness was assessed based on biopsy findings and imaging studies, such as CT and MRI. Although a TNM classification system for EMPD was proposed by Ohara et al. in 2016, it was not uniformly applied in this study, as many patients were diagnosed prior to its publication [10].

### 2.2. RT

Excluding the 30 patients who received treatment exclusively with electron beams, 51 patients underwent CT-based radiation planning, utilizing thermoformed masks or body vacuum bag systems to ensure stable positioning during the treatment. The gross tumor volume (GTV) was defined based on clinical findings and imaging results, including CT, MRI, and/or PET/CT. For 79 patients with visible skin lesions, markers were placed around the lesion by dermatologists and radiation oncologists before performing the CT scan to improve the accuracy of GTV identification. In two other patients who underwent primary lesion resection only, the GTV was defined by lymph node metastases. Among the 47 patients (58%) who received localized irradiation, the clinical target volume (CTV) was extended up to 5 cm radially from the GTV margins. For the 34 patients (42%) who underwent elective nodal irradiation, the CTV also included the elective nodal region in addition to the CTV used for localized irradiation. Elective nodal irradiation was defined as RT that included regional LN areas based on the anatomical location of the primary lesion. For lesions in the genital, perianal or inguinal region, the irradiated nodal areas were selected from the common, internal, and external iliac, presacral, obturator, and/or inguinal regions at the discretion of the treating physician. In contrast, local irradiation was defined as RT limited to the primary lesion, without prophylactic coverage of regional LN. A circumferential margin of 3–5 mm was added around the CTV to account for positional and technical uncertainties, resulting in the planning target volume (PTV). Of these patients, 51 (63%) were treated with intensity-modulated radiation therapy (IMRT) or three-dimensional conformal radiation therapy (3DCRT), while 30 patients (37%) received electron beam therapy exclusively. IMRT plans were designed to create conformal isodose distributions that minimized exposure to adjacent normal tissues, such as the bowel, sigmoid colon, rectum, and bladder. X-ray therapy was delivered using 6–10 MV beams, combined with three-dimensional treatment planning based on tumor extent and depth. Electron beam therapy was administered with a single field using 6–12 MeV electrons. In some cases, a bolus material with a thickness of 0.5–1 cm was applied to the skin surface around the gross tumor or postoperative tumor bed to ensure adequate surface dose coverage. The median radiation dose was 56 Gy, delivered over 28 fractions (30–69 Gy in 10–37 fractions). The median dose per fraction was 2.0 Gy (1.8–3.0 Gy). Three patients received a radiation dose of 30 Gy. One of them underwent palliative irradiation and was transferred to another facility shortly after completing RT. The remaining two patients were treated at the same institution in 2005–2006 with a dose of 30 Gy. The median equivalent dose in 2 Gy fractions (EQD_2_) with an α/β = 10 was 56 Gy (30–71 Gy). The median radiation dose in patients who did not undergo surgery was 60 Gy (range, 30–65 Gy), while in patients who underwent surgery, the median dose was 52.3 Gy (range, 30–69 Gy).

### 2.3. Combination Therapy

A total of 30 patients (37%) received RT as the sole treatment, while 39 patients (48%) were treated with a combination of RT and surgery. Seven patients (9%) underwent RT in conjunction with chemotherapy, and five patients (6%) received a combination of RT, surgery, and chemotherapy.

In terms of chemotherapy regimens, six patients were administered fluorouracil and cisplatin, three received docetaxel, one was treated with fluorouracil, one with etoposide, and one with tegafur–gimeracil–oteracil potassium. Out of the 44 patients (54%) who underwent surgery, 26 patients (32%) had positive surgical margins, whereas 18 patients (22%) had negative margins. Additionally, three patients (4%) received RT before surgery, and 41 patients (51%) underwent postoperative RT.

Sentinel lymph node biopsy for EMPD has only recently been covered by national insurance; therefore, it was not performed in any of the cases included in this study.

### 2.4. Follow-Up Evaluation and Statistical Analysis

In the first year following RT, clinical assessments were performed every 2 to 3 months, and CT scans of the abdomen and pelvis were conducted at intervals of at least 4 months. From the second year onwards, CT imaging was performed every 6 to 12 months. PET/CT scans were utilized when there was a suspicion of recurrence. Tumor responses were evaluated using the Response Evaluation Criteria in Solid Tumors (RECIST) [8], categorizing responses as complete response (CR), partial response (PR), progressive disease (PD), or stable disease (SD).

Local control (LC), progression-free survival (PFS), and overall survival (OS) rates were estimated using the Kaplan–Meier method. LC was defined as the proportion of patients without local recurrence after the start of RT, with local recurrence referring to tumor regrowth within the primary site or the emergence of new lesions in the genital or perianal regions. PFS was defined as the proportion of patients without any recurrence or death from the initiation of RT, with follow up ending on the date of the last event-free observation. OS was defined as the proportion of surviving patients from the start of RT, irrespective of the cause of death, until the final follow up. To assess the prognostic relevance of variables, univariate analysis was performed using the Log-rank test, and multivariate analysis was conducted with the Cox proportional hazards model. Key potential prognostic factors were selected based on prior multivariate analyses, considering patient numbers and survival outcome [6,9,10,11,12,13]. All statistical evaluations were carried out using EZR (Version 1.55; Saitama Medical Center, Jichi Medical University, Japan), a graphical interface for R (The R Foundation for Statistical Computing, Vienna, Austria) [14]. Statistical significance was defined as *p* < 0.05. RT-related adverse events were graded according to the National Cancer Institute Common Terminology Criteria for Adverse Events (NCI-CTCAE), version 5.0.

## 3. Results

### 3.1. Patient and Treatment Characteristics

Table 1 shows the characteristics of patients and treatments. Ten patients in this study had recurrent disease after surgery, while the majority had primary disease. Twenty-one patients (26%) had LN metastases, while none had distant metastases. Clinical positive lymph node metastasis was defined as lymph nodes with a short-axis diameter of ≥10 mm on CT imaging. Of these 21 patients, 7 received local irradiation, while 14 received elective nodal irradiation. Elective nodal irradiation encompassed the prophylactic regions surrounding the primary site, including the pelvic and inguinal areas. For the patient with an axillary primary tumor, irradiation was targeted to the axillary region. Sixteen patients (20%) had a tumor ≥ 10 cm, 23 (28%) had a tumor thickness ≥ 4 mm. The cutoff values for tumor diameter and thickness were determined based on previous literature [9,10].

A Chi-square test was performed to evaluate the association between chemotherapy and tumor size (<10 cm vs. ≥10 cm), tumor thickness (<4 mm vs. ≥4 mm), and LN metastasis (no vs. yes). The analysis revealed no association between chemotherapy and tumor size (*p* = 0.99) or tumor thickness (*p* = 0.73). On the other hand, chemotherapy was primarily administered to patients with LN metastasis, resulting in a significant association (*p* < 0.001) and indicating a clear bias in these cases. Additionally, the chemotherapy regimens were not standardized. Therefore, univariate and multivariate analyses regarding chemotherapy were not performed.

### 3.2. Tumor Responses

Tumor responses were not assessable in one patient (1%) due to missing data, as well as in 41 patients (51%) who had received postoperative RT. In the remaining 39 patients (48%), tumor responses were evaluated at the time of maximal tumor reduction within 1 to 3 months following RT. Among these patients, 21 (54%) achieved CR, 16 (41%) PR, and 2 (5%) exhibited PD. The response rate, calculated as the sum of CR and PR, was 95%. Both patients with PD showed progression of the primary tumor.

### 3.3. Outcomes

The median follow-up duration was 36 months (1–168 months), with a median follow-up of 38 months (2–168 months) for surviving patients. The three-year LC, PFS, and OS rates were 75% (95% confidence interval [CI]: 62–84%), 52% (95% CI: 40–63%), and 80% (95% CI: 68–88%), respectively, as illustrated in Figure 1.

At the time of the analysis, 23 patients (28%) had died, 12 (15%) from EMPD progression, 9 (11%) from unrelated causes (including four from lung cancer, hepatocellular carcinoma, cholangiocarcinoma, and rectal cancer, respectively, while the cause of death was unknown in the remaining five), and the cause of death was undetermined in 2 patients (2%). Local recurrence occurred in 18 patients (22%), with a median time to recurrence of 14 months (2–56 months) post-RT. Among the 37 patients who did not undergo surgery, 10 experienced local recurrence. Additionally, 12 patients (15%) developed regional LN metastasis, with a median time to onset of 17 months (9–35 months) following RT. Local recurrence was defined as tumor regrowth within the primary site. Regional LN metastasis was defined as disease progression within the elective nodal region. Distant metastasis was observed in 20 patients (25%), with the median time to onset being 17 months (2–50 months). The most common initial site of distant metastasis was the bone (9 cases), followed by the lungs (8 cases) and the liver (5 cases). Detailed information on initial distant metastasis sites is provided in Appendix A.

Moreover, seven patients (9%) developed secondary cancers after RT. Specific cases included one occurrence each of lung, breast, bile duct, kidney, and bladder cancer, along with two cases of colorectal cancer. The median time to secondary cancer onset was 29 months (6–134 months) post-RT.

### 3.4. Univariate Analyses

Table 2 shows the results of the log-rank test for LC, PFS, and OS. The absence of surgery was identified as a significant factor for unfavorable LC. The LC rate was lower in patients treated without surgery than in those treated with surgery (*p* = 0.04). The LC rate was slightly lower in the electron beam group than in the IMRT or 3DCRT group (*p* = 0.06).

The presence of LN metastasis and the absence of surgery were significant factors for unfavorable PFS. The PFS rate was lower in patients with LN metastasis than in those without LN metastasis (*p* = 0.005). The PFS rate was slightly lower in patients treated without surgery than in those treated with surgery (*p* = 0.051).

The presence of LN metastasis was identified as a significant factor for unfavorable OS. The OS rate was lower in patients with LN metastasis than in those without LN metastasis (*p* = 0.006). The OS rate was slightly lower in patients treated without surgery than in those treated with surgery (*p* = 0.09).

### 3.5. Multivariate Analyses

Based on the number of patients and survival events, seven major potential factors were selected for the multivariable analysis: age (<78 years vs. ≥78 years), PS (0, 1 vs. 2, 3), tumor size (<10 cm vs. ≥10 cm), tumor thickness (<4 mm vs. ≥4 mm), LN metastases (no vs. yes), surgery (yes vs. no), and RT field (elective nodal vs. local). Table 3 summarizes the results obtained in multivariate analyses of LC, PFS, and OS. The presence of LN metastasis, the absence of surgery, and the omission of elective nodal irradiation, with local irradiation only, were associated with lower LC rates (LN metastasis: hazard ratio [HR] 5.56, 95% CI 1.52–20.39, *p* = 0.01; surgery: HR 4.29, 95% CI 1.27–14.49, *p* = 0.02; RT field: HR 8.54, 95% CI 1.85–39.48, *p* = 0.006). The presence of LN metastasis, the absence of surgery, and local irradiation were associated with lower PFS rates (LN metastasis: HR 3.77, 95% CI 1.74–8.18, *p* = 0.001; surgery: HR 2.13, 95% CI 1.02–4.42, *p* = 0.04; RT field: HR 2.46, 95% CI 1.11–5.45, *p* = 0.03). The presence of LN metastasis was associated with a lower OS rate (HR 3.96, 95% CI 1.51–10.37, *p* = 0.005).

Figure 2, Figure 3 and Figure 4 show the results of univariate analyses comparing two groups for each of the seven factors found to be significant in multivariate analyses: LC differences based on the presence or absence of LN metastasis (Figure 2A), surgery (Figure 2B), and the RT field (Figure 2C); PFS differences based on LN metastasis (Figure 3A), surgery (Figure 3B), and the RT field (Figure 3C); and OS differences by LN metastasis (Figure 4).

### 3.6. Adverse Events

Two patients developed grade 2 or higher late adverse events. One patient had a grade 2 skin infection, while the other had grade 3 lymphedema. The patient with the skin infection was treated with oral antibiotics. Grade 3 lymphedema was defined as the limitation of activities of daily living. The 75-year-old female patient who developed lymphedema received RT of 45 Gy in 25 fractions (1.8 Gy per fraction) with elective nodal irradiation and 20 Gy in 10 fractions with a local irradiation boost to the vulva.

## 4. Discussion

EMPD is a malignant neoplasm that primarily affects apocrine-rich anogenital skin [1,2]. Few studies with large numbers of patients have been conducted because of the rarity of EMPD [11,12,13]. EMPD is more frequently observed in the elderly and is slightly more common in females than in males outside of Asia; however, it is reportedly more common in males in Asia [6,7,14,15,16]. In the present study, the male-to-female ratio was 42%/58% and the median age of patients was 78 years. Although the study was conducted in Asian centers, there were more females than males. When reviewing the treatment outcomes, it is important to note that some of the patients in this study were treated with a relatively low radiation dose of 30 Gy. In the present study, 30 Gy was used for three patients, including one who received palliative irradiation. For the remaining two patients, the dose was selected based on individual clinical considerations, such as advanced age or general condition, in the absence of established dosing standards at the time. This dose was administered to patients treated in 2005–2006, at a time when the optimal radiation dose for EMPD had not yet been well established. Although radiation doses varied widely (30–69 Gy), univariate analysis did not show a significant association between total dose and outcomes such as local control or overall survival. Nevertheless, given the limited sample size and variations in treatment intent, the impact of radiation dose should be interpreted with caution.

In previous studies using RT, response rates ranged from 50 to 100% HR [17,18,19,20]. The response rate in the present study was 95% (CR, 54%; PR, 41%). RT plays a central role in the definitive treatment of patients who cannot undergo surgery and in the prevention of postoperative recurrence. Brown et al. reported the outcomes of 6 patients with EMPD treated with RT [21]. Twenty-month LC and disease-free survival (DFS) rates were both 60% and the 12-month OS rate was 100%. Brierley et al. also examined the outcomes of six patients with EMPD after RT [22]. Twenty-month LC, DFS, and OS rates were all 66%. In the present study, 3-year LC, PFS, and OS rates were 78, 52, and 80%, respectively. LC and OS rates were slightly higher in the present study than previously reported values; however, PFS was slightly lower than DFS in previous studies. Since seven patients in the present study developed cancer at different sites after RT, this difference may have been more pronounced in comparisons of DFS. However, DFS was not calculated due to insufficient data. In addition, the present study included more patients than previous studies, making a direct comparison difficult. The lower PFS compared to LC and OS in the present study may be partly explained by the relatively high incidence of distant metastases, which occurred in 25% of patients. In addition, several patients experienced regional lymph node recurrence or developed second primary cancers during the follow-up period. While these events contributed to a decrease in PFS, many patients were able to receive salvage treatment or had stable disease, which may have helped maintain overall survival. These findings highlight the importance of appropriate follow up and may suggest a potential role for systemic therapy in selected patients. Among the seven patients who developed secondary malignancies, none of the tumors arose within the irradiated field. Given the advanced age of the cohort and the long latency periods observed, these cases are more likely attributable to pre-existing risk factors rather than to RT itself.

Multivariate analyses confirmed that the presence of LN metastasis, the absence of surgery, and local irradiation significantly reduced PFS rates. The prognostic factors of the presence of LN metastasis and the absence of surgery were consistent with previous findings [6,23,24,25]. The primary treatment for EMPD is considered to be surgery [7,17,26]. In our multivariate analysis, LC rates were significantly lower in the group without surgery than in the surgery group. However, since many patients with EMPD are elderly, radical complete resection is often not feasible when the tumor is too large. In previous studies, many cases of EMPD were treated with electron beams [22,27]. Nevertheless, since it is difficult to irradiate large or deep-seated targets, including LN metastases, with electron beams, the use of techniques such as 3DCRT and IMRT has increased in recent years [17,23,28]. Although the application of IMRT for EMPD has not yet been described, its utility in increasing the dose to the target while reducing that to organs at risk has been suggested [29,30]. In the univariate analysis, LC rates were slightly lower with RT using electron beams alone than with IMRT or 3DCRT (*p* = 0.06). This trend may be due to the difficulty of delivering a sufficiently uniform dose to the target with electron beams. Hata et al. identified LN metastasis as a prognostic factor for EMPD and highlighted the usefulness of elective nodal irradiation for patients with LN metastasis [23]. Elective nodal irradiation with doses of 41.4–50.4 Gy may eradicate microscopic LN metastases and effectively prevent recurrence. In the present study, local irradiation significantly worsened both LC and PFS compared to elective nodal irradiation. Based on our findings, we suggest that elective nodal irradiation may offer better outcomes than local irradiation in EMPD treatment. To select the optimal radiation dose for the elective nodal field in EMPD, further studies with more patients are warranted. Additionally, previous studies identified tumor size and LN metastasis as prognostic factors in EMPD [6,9,31]. While our results for LN metastasis were consistent with previous studies, tumor size did not emerge as a significant prognostic factor in either univariate or multivariate analysis. This discrepancy may be attributed to the study’s retrospective design, as tumor size could only be categorized as ≥10 cm or <10 cm for many patients. To obtain more accurate results, prospective studies with precise tumor size measurements are necessary. Differentiating between local recurrence and regional LN metastasis remains difficult, especially in cases where the primary lesion and metastatic LN are anatomically contiguous. Further prospective studies incorporating standardized imaging criteria and pathological confirmation would be valuable in improving classification accuracy.

Patients with EMPD often develop distant metastases to the lungs, liver, and bones [17,32,33]. In the present study, 20 out of 81 patients (25%) developed distant metastases after RT and the most common organs for distant metastases were the lungs and bones. Systemic therapy may be necessary to prevent distant metastasis of EMPD; however, an effective chemotherapy regimen has yet to be established for EMPD [34,35,36]. In present study, the number of patients who received chemotherapy was small, and it was predominantly administered to patients with lymph node metastasis, resulting in a substantial selection bias. The chemotherapy regimens used were heterogeneous, and due to the limited number of patients and the retrospective nature of the study, it was not possible to assess the precise impact of chemotherapy on treatment outcomes. Moreover, the response to chemotherapy, especially in those with LN metastasis, was not consistently documented or robust enough to draw meaningful conclusions. Therefore, chemotherapy was excluded from statistical analyses. Although the role of chemotherapy in EMPD remains unclear, given the relatively high incidence of distant metastases observed in this study, further investigation into the efficacy of systemic therapy is crucial. A more comprehensive analysis, incorporating a larger number of cases with uniform chemotherapy data, could help to elucidate the potential benefits of chemotherapy, including its impact on survival rates, disease progression, and quality of life. A more detailed evaluation of clinical indications, treatment response, and long-term outcomes may provide valuable insights for future treatment strategies.

We encountered grade 2 skin infection and grade 3 lymphedema as late adverse events. In RT to the genital and perianal regions, radiation dermatitis is almost inevitable, making proper skin care essential to prevent bacterial infections. The patient who developed lymphedema had a 12 cm tumor in the vulvar region and received RT of 45 Gy in 25 fractions with elective nodal irradiation and 20 Gy in 10 fractions with a local irradiation boost to the vulva. The extensive irradiation field may have contributed to the onset of lymphedema. However, as this was a retrospective, multi-institutional study, the documentation of late toxicities was not consistent across participating centers. In many cases, detailed records regarding late adverse events were unavailable. Although techniques such as IMRT and the use of bolus material likely contributed to minimizing toxicities, the potential for underreporting—stemming from variability in documentation—should be acknowledged. To better characterize the long-term effects of RT in patients with EMPD, more consistent and detailed toxicity assessment will be essential in future studies.

The present study has several limitations inherent to a retrospective design. There were three patients whose PS was unknown due to inadequate evaluations, which may be insufficient to accurately assess survival rates. Furthermore, some patients were not followed up at the prescribed intervals, resulting in missing data at various time points. Moreover, since this was a multi-center collaborative study, the evaluation of tumor responses and adverse events requiring examinations may have been inconsistent. Furthermore, because of the retrospective design, tumor size and thickness were often assessed only dichotomously, limiting a more detailed analysis of their prognostic impact. Prospective studies with continuous or more granular measurements are warranted to better clarify their roles. In consideration of these limitations, the present results need to be interpreted with caution.

## 5. Conclusions

The present study identified prognostic factors for EMPD after RT, suggesting that the absence of surgery and the omission of elective nodal irradiation may negatively affect the outcomes. Given the rarity of this disease and its higher prevalence among elderly patients, especially those with LN metastasis, prospective studies are needed to establish optimal treatment strategies.

## Figures and Tables

**Figure 1 cancers-17-01507-f001:**
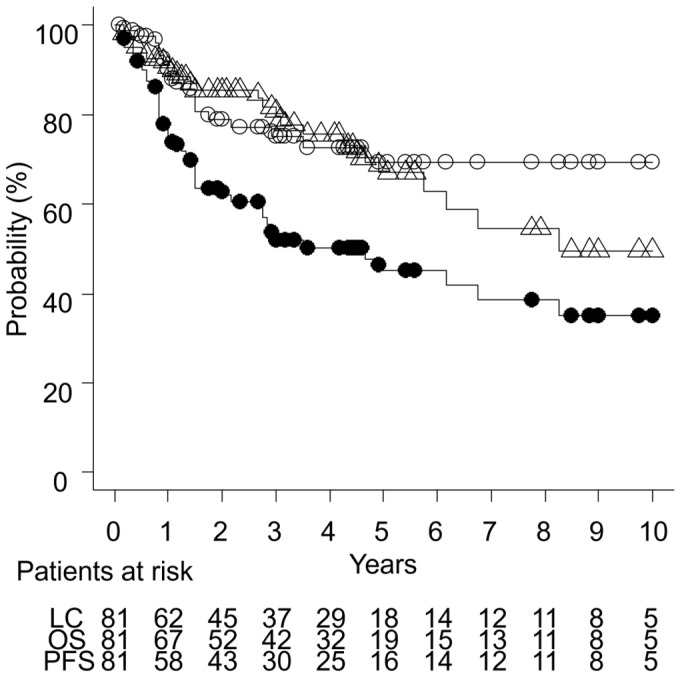
Curves of local control (LC, open circle), progression-free survival (PFS, filled circle), and overall survival (OS, triangle) for 81 patients with extramammary Paget’s disease.

**Figure 2 cancers-17-01507-f002:**
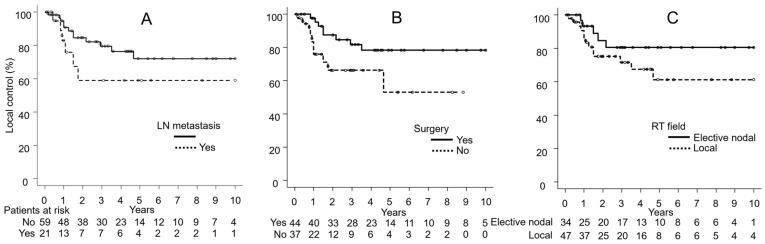
(**A**) Differences in local control by the presence or absence of lymph node (LN) metastasis, (**B**) the use of surgery, and (**C**) the radiation therapy (RT) field.

**Figure 3 cancers-17-01507-f003:**
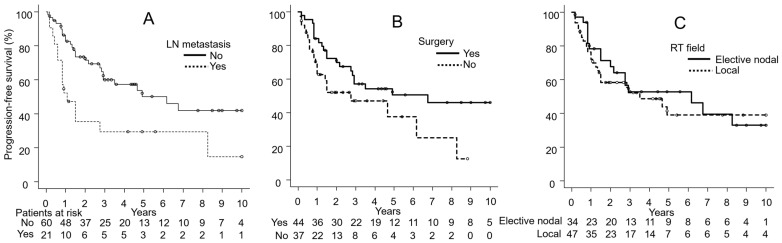
(**A**) Differences in progression-free survival by the presence or absence of lymph node (LN) metastasis, (**B**) the use of surgery, and (**C**) the radiation therapy (RT) field.

**Figure 4 cancers-17-01507-f004:**
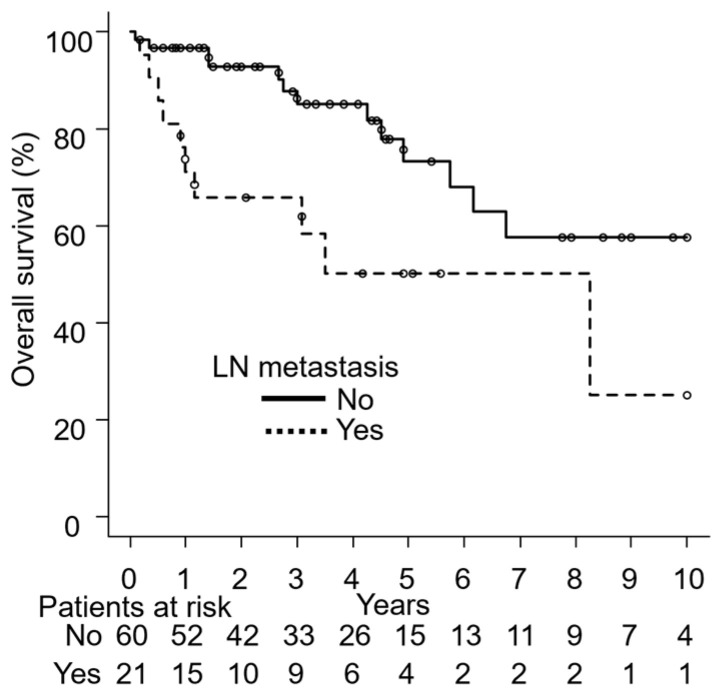
Differences in overall survival by the presence or absence of lymph node (LN) metastasis.

**Table 1 cancers-17-01507-t001:** Patient and treatment characteristics.

Characteristic	*n* = 81
Age (years)	78 (50–95)
<78/≥78	35 (43%)/46 (57%)
Sex	
Male/female	34 (42%)/47 (58%)
PS	
0/1/2/3/missing	27 (33%)/35 (43%)/14 (17%)/2 (2%)/3 (4%)
Primary tumor site	
Genital/perianal/inguinal/axillary/genital to perianal	71 (88%)/4 (5%)/1 (1%)/1 (1%)/4 (5%)
Tumor size (cm)	6 (0.6–20)
<10/≥10	65 (80%)/16 (20%)
Tumor thickness (mm)	
<4/≥4	58 (72%)/23 (28%)
Lymph node metastases	
No/yes	60 (74%)/21 (26%)
Treatment methods	
RT	30 (37%)
RT + surgery	39 (48%)
RT + chemotherapy *	7 (9%)
RT + surgery + chemotherapy	5 (6%)
RT field	
Local/elective nodal	47 (58%)/34 (42%)
RT methods	
IMRT or 3DCRT/electron beams alone	51 (63%)/30 (37%)
Total dose (Gy)	56 (30–69)
EQD_2_ (Gy)	56 (30–71)

PS = performance status, RT = radiation therapy, IMRT = intensity-modulated radiation therapy, 3DCRT = three-dimensional conformal radiation therapy, EQD_2_ = equivalent dose in 2 Gy fractions with α/β  =  10 Gy, *: The most common schedule is weekly dosing for 3 weeks followed by a 1-week interval.

**Table 2 cancers-17-01507-t002:** Results of log-rank tests for potential variables affecting each outcome.

Characteristic	Variable	*n*	Local Control		Progression-Free Survival		Overall Survival	
			3-Year Rate (%)	*p*-Value	3-Year Rate (%)	*p*-Value	3-Year Rate (%)	*p*-Value
Age (years)	<78	35	79	0.46	57	0.18	84	0.20
	≥78	46	72		48		75	
Sex	Female	47	78	0.42	58	0.62	77	0.55
	Male	34	71		44		84	
PS	0, 1	62	78	0.64	53	0.45	83	0.73
	≥2	16	85		62		63	
Tumor size (cm)	<10	67	75	0.59	51	0.86	79	0.42
	≥10	14	75		56		85	
Tumor thickness (mm)	<4	58	79	0.63	58	0.31	80	0.92
	≥4	23	64		38		80	
Lymph node metastasis	Yes	21	59	0.18	30	0.005	66	0.006
	No	60	80		60		85	
Surgery	Yes	44	82	0.04	57	0.051	85	0.09
	No	37	66		47		72	
RT field	Local	47	72	0.18	52	0.62	77	0.89
	Elective nodal	34	80		53		83	
RT methods	IMRT or 3DCRT	51	83	0.06	51	0.35	86	0.18
	Electron beams only	30	65		55		76	
EQD_2_ (Gy)	≥56	45	81	0.20	52	0.56	77	0.13
	<56	36	68		52		84	

PS = performance status, RT = radiation therapy, IMRT = intensity-modulated radiation therapy, 3DCRT = three-dimensional conformal radiation therapy, EQD_2_ = equivalent dose in 2 Gy fractions with α/β  =  10 Gy.

**Table 3 cancers-17-01507-t003:** Multivariate analysis of major potential variables affecting survival.

Variable	Local Control			Progression-Free Survival			Overall Survival		
	HR	95% CI	*p*-Value	HR	95% CI	*p*-Value	HR	95% CI	*p*-Value
Age (years)									
0: <78	0.86	0.28–2.66	0.80	1.09	0.54–2.23	0.81	1.40	0.56–3.51	0.47
1: ≥78									
PS									
0:0, 1	0.85	0.27–2.63	0.77	0.62	0.31–1.23	0.17	0.57	0.24–1.38	0.22
1:2-									
Tumor size (cm)									
0:<10	2.58	0.69–9.65	0.16	1.49	0.63–3.53	0.37	0.85	0.23–3.14	0.81
1: ≥10									
Tumor thickness (mm)									
0: <4	2.45	0.75–8.03	0.14	1.57	0.77–3.21	0.22	1.21	0.47–3.11	0.69
1: ≥4									
Lymph node metastasis									
0: No	5.56	1.52–20.39	0.01	3.77	1.74–8.18	0.001	3.96	1.51–10.37	0.005
1: Yes									
Surgery									
0: Yes	4.29	1.27–14.49	0.02	2.13	1.02–4.42	0.04	1.83	0.72–4.65	0.20
1: No									
RT field									
0: Elective nodal	8.54	1.85–39.48	0.006	2.46	1.11–5.45	0.03	1.77	0.66–4.77	0.26
1: Local									

HR = hazard ratio, 95% CI = 95% confidence interval, PS = performance status, RT = radiation therapy.

## Data Availability

The data can be shared upon request.

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
