# Peer review of "Clinical Outcomes and Prognostic Factors for Extramammary Paget’s Disease Treated with Radiation Therapy: A Multi-Institutional Observational Study"

_cancers, 2025, doi:10.3390/cancers17091507_

Round 1

Reviewer 1 Report

Comments and Suggestions for Authors

Dear Dr. Niwa and colleagues,

We read your study on the outcomes of radiation therapy in Extramammary Paget’s Disease (EMPD) with great interest. We commend you for assembling a large, multi-institutional dataset, which provides a valuable contribution in a field that remains underexplored due to the rarity of this disease. Your findings reinforce established knowledge regarding the prognostic importance of lymph node metastases, as well as the role of surgery and elective nodal irradiation.

We would, however, like to raise two points for discussion:

1. Tumor size and thickness thresholds.

The use of 10 cm as a cut-off for tumor size and 4 mm for tumor thickness appears somewhat arbitrary. While these thresholds have been cited in previous literature, a more robust justification—such as ROC analysis or stratified survival curves—would have strengthened their validity. Furthermore, given the often multifocal and heterogeneous nature of EMPD, we believe that future studies should aim to develop more standardized and detailed approaches for tumor measurement and risk stratification.

2. Limited analysis of the role of chemotherapy.

Although chemotherapy is described in detail in terms of regimens and patient distribution, it was excluded from statistical analyses due to heterogeneity and selection bias (primarily administered to patients with nodal metastases). This limits the ability to assess its potential systemic impact, especially in a disease where distant metastases occurred in 25% of patients. A more in-depth discussion of clinical indications and observed outcomes could still offer useful insights for clinical practice and the design of future studies.

Despite these limitations, we believe your work represents a contribution to the literature and strengthens the evidence supporting radiation-based approaches for EMPD, particularly in elderly or inoperable patients.

3. I believe there is an error in Reference 18, which includes the unrelated phrase “0 Clinical Original Contribution” and an incorrect journal abbreviation (“Radiarmn Oncol. Biol Phys”).

Author Response

We sincerely appreciate your thoughtful and encouraging comments on our manuscript. Your recognition of the significance of assembling a large, multi-institutional dataset and of our findings regarding lymph node metastasis, surgery, and elective nodal irradiation is truly encouraging. We have carefully considered the two points you raised and provide our responses below.

Comments 1: Tumor size and thickness thresholds.

The use of 10 cm as a cut-off for tumor size and 4 mm for tumor thickness appears somewhat arbitrary. While these thresholds have been cited in previous literature, a more robust justification—such as ROC analysis or stratified survival curves—would have strengthened their validity. Furthermore, given the often multifocal and heterogeneous nature of EMPD, we believe that future studies should aim to develop more standardized and detailed approaches for tumor measurement and risk stratification.

Response 1: We appreciate the reviewer’s insightful comment. As noted in the Discussion, tumor size was not identified as a significant prognostic factor in our analysis, which may be due to limitations in the available data. Because of the retrospective nature of the study, tumor size was frequently documented only as “≥10 cm” or “<10 cm,” which precluded more granular analysis. Similarly, data on tumor thickness were available for only a limited number of patients, and the 4 mm threshold was selected based on previous reports rather than validation within our own study. We have added this limitation to the Discussion (page 13, third paragraph, line 412-418) and emphasized the need for prospective studies with standardized and detailed tumor measurements.

Comments 2. Limited analysis of the role of chemotherapy.

Although chemotherapy is described in detail in terms of regimens and patient distribution, it was excluded from statistical analyses due to heterogeneity and selection bias (primarily administered to patients with nodal metastases). This limits the ability to assess its potential systemic impact, especially in a disease where distant metastases occurred in 25% of patients. A more in-depth discussion of clinical indications and observed outcomes could still offer useful insights for clinical practice and the design of future studies.

Despite these limitations, we believe your work represents a contribution to the literature and strengthens the evidence supporting radiation-based approaches for EMPD, particularly in elderly or inoperable patients.

Response 2: Thank you for your valuable comment regarding the limited analysis of chemotherapy. As noted, chemotherapy was predominantly administered to patients with nodal metastases, leading to considerable selection bias and heterogeneity. For reference, below is a survival curve of PFS for the group of patients who received chemotherapy and the group of patients who did not receive chemotherapy. We did not include this figure because readers might be misled into believing that chemotherapy would increase the incidence of relapse. For this reason, we did not include it in the statistical analyses. However, we agree that further discussion on clinical indications and observed outcomes may be helpful. We have revised the Discussion section (page 12, second paragraph, line 379-394) to better address the potential role of chemotherapy and the need for further studies to evaluate its efficacy in the management of EMPD.

Comments 3: I believe there is an error in Reference 18, which includes the unrelated phrase “0 Clinical Original Contribution” and an incorrect journal abbreviation (“Radiarmn Oncol. Biol Phys”).

Response 3: Thank you for pointing out the error in Reference 18. We have removed the unrelated phrase “0” and corrected the journal abbreviation to “Int J Radiat Oncol Biol Phys” as appropriate.

Reviewer 2 Report

Comments and Suggestions for Authors

This is a well-conducted and valuable multi-institutional retrospective study on a rare condition—Extramammary Paget’s disease (EMPD)—providing insights into the outcomes and prognostic factors associated with radiation therapy (RT). The paper is clearly written, methodologically sound, and presents relevant findings that could inform future prospective trials. The study benefits from a relatively large sample size given the rarity of EMPD.

However, some aspects require clarification, expansion, or slight revision for improved clarity and scientific rigor.

Major Comments

  1. Justification of RT Dose Range:

    • Comment: The radiation doses range from 30 Gy to 69 Gy, with some patients receiving very low doses.

    • Recommendation: Please provide a stronger rationale for this wide range and clearly state how the dose differences might have affected the outcomes (e.g., why 30 Gy was considered acceptable in some cases).

  2. Chemotherapy Data Use:

    • Comment: Chemotherapy was not included in multivariate analysis due to heterogeneity and bias.

    • Recommendation: Consider including a brief analysis or even a descriptive summary in the Discussion regarding how chemotherapy may have influenced the subgroup with LN metastasis, and clarify the limitation more explicitly.

  3. Definition of Elective Nodal Irradiation:

    • Comment: The study refers to “elective nodal irradiation” vs. “local irradiation.”

    • Recommendation: Clearly define these terms earlier in the methods section, preferably with a schematic figure or table showing the irradiated areas.

  4. Survival Metrics Interpretation:

    • Comment: The differences in LC, PFS, and OS are reported, but not deeply discussed.

    • Recommendation: Provide a more nuanced interpretation of why PFS is notably lower than LC and OS (e.g., role of distant metastases, second malignancies, or surveillance intensity).

  5. Assessment of Late Adverse Events:

    • Comment: Only two adverse events (Grade ≥2) are reported.

    • Recommendation: Please clarify whether this low number reflects underreporting or a genuine absence, especially given the wide radiation fields in some patients.

Minor Comments

  1. Grammar and Style:

    • Some sections have typographical errors (e.g., “pa- get’s” on page 1).

    • Recommendation: Please conduct a careful language edit throughout the manuscript to correct spacing and formatting issues.

  2. Figures and Tables:

    • Figure 2 is helpful but could be clearer.

    • Recommendation: Consider combining some survival curves into a single multi-panel figure and improving label readability.

  3. Secondary Malignancies:

    • Comment: Seven patients developed secondary cancers post-RT.

    • Recommendation: Please clarify whether these are potentially related to RT or pre-existing risk factors (age, comorbidities).

  4. TNM Staging Reference:

    • The proposed TNM system by Ohara et al. is briefly mentioned in the references.

    • Recommendation: Consider referencing this system in the staging description section, especially if it was used in patient classification.

  5. Conclusion Section:

    • The conclusion is appropriate but slightly repetitive.

    • Recommendation: Summarize the main implications more succinctly and clearly highlight the need for prospective studies.

Author Response

We appreciate the reviewers’ pertinent suggestions and their efforts in reviewing our paper. We have carefully read the comments of the reviewers, and have modified our manuscript according to their suggestions. The changes are highlighted in red. We hope that you will find this revised manuscript acceptable for publication in Cancers, and we look forward to your response.

Major Comments

Comments 1: Justification of RT Dose Range:

Comment: The radiation doses range from 30 Gy to 69 Gy, with some patients receiving very low doses.

Recommendation: Please provide a stronger rationale for this wide range and clearly state how the dose differences might have affected the outcomes (e.g., why 30 Gy was considered acceptable in some cases).

Response 1: Thank you for the comment. We have revised the Discussion to clarify the reasons for the wide dose range (page 11, first paragraph, line 308-317). Three patients received 30 Gy, including one palliative case and two treated in 2005–2006, before optimal dosing was established. We also noted that in our univariate analysis, radiation dose was not significantly associated with outcomes, though this should be interpreted with caution due to the small number of low-dose cases.

Comments 2: Chemotherapy Data Use:

Comment: Chemotherapy was not included in multivariate analysis due to heterogeneity and bias.

Recommendation: Consider including a brief analysis or even a descriptive summary in the Discussion regarding how chemotherapy may have influenced the subgroup with LN metastasis, and clarify the limitation more explicitly.

Response 2: We thank the reviewer for this insightful comment. In response, we have expanded the Discussion section (page 12, second paragraph, line 379-394) to include a more detailed description of the chemotherapy data, particularly regarding its use in patients with lymph node metastasis. For reference, below is a survival curve of PFS for the group of patients who received chemotherapy and the group of patients who did not receive chemotherapy. We did not include this figure because readers might be misled into believing that chemotherapy would increase the incidence of relapse. We clarified that the small number of patients and the heterogeneity in regimens precluded statistical analysis, and we discussed how these limitations affect interpretation of the findings. We also emphasized the need for further investigation into the role of systemic therapy in EMPD and the importance of accumulating standardized data to inform future treatment strategies.

Comments 3: Definition of Elective Nodal Irradiation:

Comment: The study refers to “elective nodal irradiation” vs. “local irradiation.”

Recommendation: Clearly define these terms earlier in the methods section, preferably with a schematic figure or table showing the irradiated areas.

Response 3: Thank you for your helpful suggestion. We have now added a clear definition of “elective nodal irradiation” and “local irradiation” in the Methods section (page 3, third paragraph, line 116-122). Because the nodal target areas vary depending on the anatomical location of the primary lesion or lymph nodes, we did not include a schematic figure or table, as we believe a textual explanation is more appropriate in this context.

Comments 4: Survival Metrics Interpretation:

Comment: The differences in LC, PFS, and OS are reported, but not deeply discussed.

Recommendation: Provide a more nuanced interpretation of why PFS is notably lower than LC and OS (e.g., role of distant metastases, second malignancies, or surveillance intensity).

Response 4: Thank you for pointing this out. As suggested, we have added a more detailed interpretation in the Discussion (page 11, second paragraph, line 331-338) to explain why PFS was notably lower than LC and OS. We now mention the roles of distant metastases, regional recurrence, and second primary cancers, as well as the influence of salvage treatments on maintaining overall survival despite disease progression.

Comments 5: Assessment of Late Adverse Events:

Comment: Only two adverse events (Grade ≥2) are reported.

Recommendation: Please clarify whether this low number reflects underreporting or a genuine absence, especially given the wide radiation fields in some patients.

Response 5: Thank you for your valuable comment. We agree that the number of reported late adverse events appears to be low. As this was a retrospective, multi-institutional study, detailed documentation of late toxicities varied across facilities, and in some cases, specific information on late adverse events was unavailable. Therefore, the reported incidence may underestimate the true frequency of late toxicity. We have clarified this limitation in the revised Discussion section (page 13, second paragraph, line 401-407).

Minor Comments

Comments 1: Grammar and Style:

Some sections have typographical errors (e.g., “pa- get’s” on page 1).

Recommendation: Please conduct a careful language edit throughout the manuscript to correct spacing and formatting issues.

Response 1: Thank you for pointing this out. The issue such as “pa-get’s” seems to result from automatic hyphenation when using the journal-provided Word template. Nevertheless, we have carefully reviewed the manuscript and corrected any typographical or formatting issues we could identify.

Comments 2: Figures and Tables:

Figure 2 is helpful but could be clearer.

Recommendation: Consider combining some survival curves into a single multi-panel figure and improving label readability.

Response 2: We appreciate the reviewer’s helpful comment. In response, we have divided the original Figure 2 into three separate multi-panel figures (Figs. 2A–C for LC, 3A–C for PFS, and 4A for OS) to improve clarity and readability. Each panel now presents Kaplan–Meier survival curves comparing two groups for each significant factor identified in the multivariate analyses. Additionally, we have adjusted the axis labels and text size to enhance legibility.

Comments 3: Secondary Malignancies:

Comment: Seven patients developed secondary cancers post-RT.

Recommendation: Please clarify whether these are potentially related to RT or pre-existing risk factors (age, comorbidities).

Response 3: We appreciate the reviewer’s comment. All secondary malignancies were diagnosed several years after RT, and their locations were not within the irradiated fields. Given the patients’ advanced age and potential comorbidities, these malignancies are more likely attributable to underlying risk factors rather than RT itself. We have added a clarification to the Results and Discussion sections accordingly (page 11, second paragraph, line 338-341).

Comments 4: TNM Staging Reference:

The proposed TNM system by Ohara et al. is briefly mentioned in the references.

Recommendation: Consider referencing this system in the staging description section, especially if it was used in patient classification.

Response 4: We appreciate the reviewer’s comment regarding the TNM staging system proposed by Ohara et al. As many patients in our study were diagnosed before the publication of this classification in 2016, it was not uniformly applied across all cases. We have clarified this point in the Methods section of the manuscript (page 3, second paragraph, line 101-103).

Comments 5: Conclusion Section:

The conclusion is appropriate but slightly repetitive.

Recommendation: Summarize the main implications more succinctly and clearly highlight the need for prospective studies.

Response 5: We appreciate the reviewer’s suggestion. The Conclusion section has been revised to summarize the findings more succinctly and to clearly highlight the need for prospective studies, particularly in elderly patients with poor prognostic factors (page 13, fourth paragraph, line 420-424).
